# Highly defective ultra-small tetravalent MOF nanocrystals

Shan Dai [1,2], Charlotte Simms [3], Gilles Patriarche[4], Marco Daturi [2], Antoine Tissot [1] ✉, Tatjana N. Parac-Vogt [3] ✉ & Christian Serre [1] ✉

The size and defects in crystalline inorganic materials are of importance in many applications, particularly catalysis, as it often results in enhanced/ emerging properties. So far, applying the strategy of modulation chemistry has been unable to afford high-quality functional Metal–Organic Frameworks (MOFs) nanocrystals with minimized size while exhibiting maximized defects. We report here a general sustainable strategy for the design of highly defective and ultra-small tetravalent MOFs (Zr, Hf) crystals (ca. 35% missing linker, 4–6 nm). Advanced characterizations have been performed to shed light on the main factors governing the crystallization mechanism and to identify the nature of the defects. The ultra-small nanoMOFs showed exceptional performance in peptide hydrolysis reaction, including high reactivity, selectivity, diffusion, stability, and show emerging tailorable reactivity and selectivity towards peptide bond formation simply by changing the reaction solvent. Therefore, these highly defective ultra-small M(IV)-MOFs particles open new perspectives for the development of heterogeneous MOF catalysts with dual functions.

Over the past few decades, the development of colloidal nanocrystals has led to a revolution in material science due to their very appealing properties in heterogeneous catalysis, optics, biology, and engineering[1,2]. Indeed, most nanomaterials undergo dramatic changes in their properties when their particle size lies in the ultra-small scale (e.g., below 5–10 nm) such as the quantum size effect in semiconductor materials[3], catalytic properties for inert noble metals[4], or electrical conductivity for insulators[5]. Metal–organic framework nanocrystals (nanoMOFs) are porous solids assembled from metal ions/oxoclusters and organic linkers[6,7]. The reduction of the MOFs size to the nanoscale has imparted nanoMOFs with various enhanced properties (i.e., catalysis, sensing, biomedicine…)[8–11] and novel features (flexibility, optical properties)[8,12,13], but despite advances, the design of ultra-small MOF nanoparticles still faces severe difficulties[14]. This can be due either to the much larger unit-cell parameters of MOFs in comparison with inorganic nanomaterials and/or to the limited chemical stability of targeted nanoMOFs. However, at this ultra-small size, the majority of the atoms of MOFs lie close to the external surface, decorated with larger cavities than the constitutive inner ones. This maximizes the interface for substrate interaction alongside largely decreased diffusion/desorption path length[4], naturally resulting in enhanced catalytic properties.

Defect engineering is a long term interest in crystalline nanomaterials, particularly due to the influence of vacant sites on catalysis[15,16]. Interestingly, structural defects in MOFs have shown similar optimizations towards catalytic properties and/or gas separation[17]. However, this is in most cases associated with a lower chemical stability due to the reduction of metal-ligand connectivity and/or the presence of additional accessible metal sites[18]. UiO-66(Zr) or $Zr_6O_4(OH)_4(BDC)_6$ (BDC = Benzene-1,4-dicarboxylic acid) is a prototypical zirconium-based MOF with an excellent thermal and chemical stability due to its high metal-ligand connectivity (12-connected mode) and robust Zr-

[1]Institut des Matériaux Poreux de Paris, Ecole Normale Supérieure, ESPCI Paris, CNRS, PSL University, 75005 Paris, France. [2]Normandie Université, ENSICAEN, UNICAEN, CNRS, Laboratoire Catalyse et Spectrochimie, 14000 Caen, France. [3]Laboratory of Bioinorganic Chemistry, Department of Chemistry, KU Leuven, Celestijnenlaan 200F, 3001 Leuven, Belgium. [4]Université Paris-Saclay, CNRS, Centre de Nanosciences et de Nanotechnologies, 91120 Palaiseau, France. ✉e-mail: antoine.tissot@ens.psl.eu; tatjana.vogt@kuleuven.be; christian.serre@ens.psl.eu

carboxylate bonds[19]. Consequently, defect engineering in MOFs has been to date mainly focused on UiO-66(Zr) and its derivatives[18]. The most typical method for the defect engineering in Zr-MOFs is the modulator-induced-defect approach (MIDA), where a mono-carboxylate modulator (formate, acetate…) is added, binding preferentially with metal centers in place of the linker, leading to the missing ligand vacancy defects (Fig. 1a). As such, the defect content can be controlled by the amount of the modulator used[20–23].

During the MOF synthesis, the modulator binds to the $Zr_6$ nodes to produce crystals with lower connectivity, and consequently with larger size due to lower nucleation rate and crystal growth kinetics. Although this is a reliable way to produce MOF particles with tunable particle sizes[24,25], it is at the expense of control over the number of defects (Fig. 1a)[26–29]. The MIDA strategy therefore prevents the synthesis of highly defective ultra-small nanoMOFs which are optimal candidates for catalysis. Notably, numerous reports have pointed out the importance of overcoming diffusion barriers, with MOF catalyzed reactions mainly taking place on the outer surface of the particles[30–32]. This is the case particularly when the size of the substrate is comparable to the aperture size of the MOF's pores. Although the substrates may partially diffuse inside the MOF framework, the desorption of the resulting products can be hampered by kinetic limitations. Relying on exfoliated high aspect ratio 2D porous nanosheets is an appealing alternative strategy to overcome these limitations[33]. However, these nanomaterials are usually more challenging to prepare (or exfoliate) and/or exhibit usually a reduced stability compared with their related 3D counterparts. The MIDA approach, usually carried out in toxic N,N-dimethylformamide (DMF) at high temperature and pressure, has mainly been explored with UiO-66 with missing linker defect content of ca. 10–20%[18,34,35]. Additionally, it has not been extended to the functionalized derivatives and thus, the full potential of this defective nanoMOFs family has also not been exploited so far. Therefore, developing new versatile routes to both downsize robust nanoMOFs, whilst ensuring a high defect content and maintaining reasonable thermal/chemical stability of the ultra-small MOF nanoparticles to address challenging catalytic reactions is a key challenge to overcome.

We report here a sustainable route to produce a series of ultra-small UiO-type MOFs (4–6 nm) with exceptionally high defect content. A set of advanced characterization techniques revealed that the large defect content on such small UiO-66 nanoparticles is attributed to the presence of missing linker defects and that the crystallization is growth-dominated. Noteworthy, this strategy is versatile and can be applied to many UiO-66(Zr)-X derivatives (X = $NH_2$, $NO_2$, $(OH)_2$, Br), to the Hf counterpart UiO-66(Hf) and finally to other Zr-MOFs structures like the Zr fumarate MOF-801(Zr), resulting in ultra-small nanoMOFs with very high defect content. Additionally, our mild green synthetic conditions are far more sustainable than the traditional solvothermal routes, which is of interest to save energy and/or strongly reduce the quantity of hazardous wastes. Moreover, the nanoMOFs synthesized here present excellent catalytic performance in peptide bond hydrolysis showing much better reactivity, chemical diffusion, selectivity and stability than benchmark catalysts. Significantly, these nanoMOFs show bifunctionality as by simply changing the reaction solvent, the hydrolysis of peptide bonds can be replaced by the opposite condensation reaction, resulting in amide bond formation. Additionally, these nanoMOFs also show tailorable selectivity due to the molecular-sieving effect.

## Results and discussion
### Materials synthesis and characterizations

To prepare high quality ultra-small nanoparticles of UiO-66(Zr) with high defect content, we first considered carefully the main relevant state-of-the-art strategies. For instance, the acidity of the solution was shown to significantly influence the kinetics of crystallization due to the changes in protonation state of the carboxylic acids that lead to faster kinetics at higher pH[24,36,37]. In addition, the presence of water in the reaction mixture appeared to be a critical factor in determining the defects resulting from the formation of Zr-OH or $Zr-OH_2$ bonds rather than Zr-ligand connections[38]. Using a low synthesis temperature was also shown to be beneficial towards both the defect engineering[35] and downsizing due to the inhibited formation of coordination bonds and to limited Ostwald ripening (illustrated in Fig. 1b)[39]. Thus, to achieve our ambitious goal to prepare ultra-small nanoMOFs with a high defect content, we developed a simple green strategy that: (i) avoids the use of very acidic Zr salts (e.g., $ZrCl_4$, $ZrOCl_2 \cdot xH_2O$) and slightly acidic modulators (e.g., formic acid) by using pre-synthesized $Zr_6$ acetate oxoclusters (Supplementary Fig. 1); (ii) discards DMF and replaces it by water and ethanol to avoid the release of formates upon DMF degradation (and enables a sustainable approach); (iii) ensures a faster dissolution of the organic linker by diluting the reaction media in ethanol, subsequently accelerating the synthesis kinetics; and (iv) is operated at room temperature, which is energy saving.

The initial synthesis of UiO-66(Zr) was performed by first mixing acetate capped $Zr_6$ oxoclusters with acetic acid. Water, ethanol, and benzene-1,4-dicarboxylic acid (BDC) were subsequently introduced in the oxocluster solution (see detail in SI). After 2 h at room

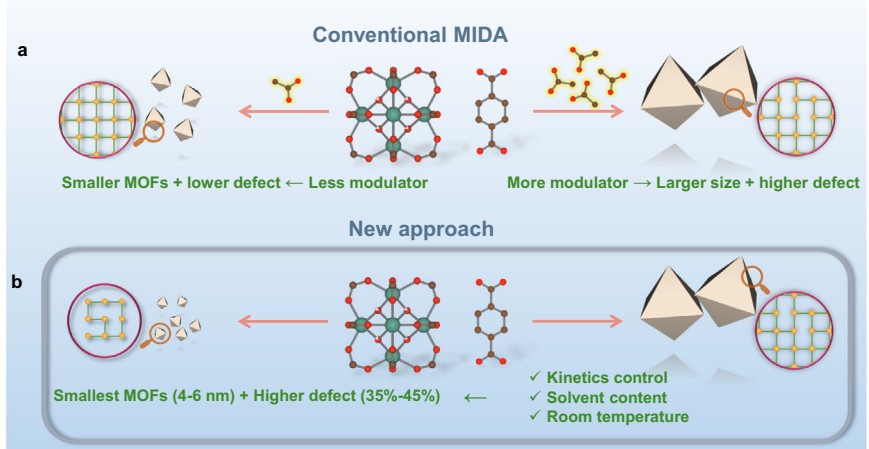

**Fig. 1 | Comparison of MOF nanoparticles synthesis approaches.** Scheme of (**a**) the conventional modulator-induced-defect approach (MIDA) for the size-defect tuning of MOFs, (**b**) our approach that produces ultra-small and highly defective tetravalent-MOFs nanoparticles, Red, green, and brown represent O, Metal(IV), and C atoms, respectively; yellow sphere and green lines within the purple circles indicate the metal nodes and organic ligand, respectively.

temperature under stirring, the resulting solid showed a PXRD pattern (see Supplementary Fig. 2) in agreement with the theoretical diffraction pattern of UiO-66. Transmission Electron Microscope (TEM) indicated a particle size of 40 nm (±7) (Supplementary Fig. 3). This value is close to the particle size (44 nm) calculated from Scherrer equation, indicating that the particles are mainly single crystal domains. The 77 K $N_2$ adsorption (Supplementary Fig. 4) showed a type I isotherm with extremely high $N_2$ capacity (404 cm$^3$/g) and a calculated Brunauer−Emmett−Teller (BET) surface area (1617 (±5) m$^2$/g), larger than the BET surface area of defect-free UiO-66(Zr) (-1000 m$^2$/g). Such a huge expansion in the surface area is indicative of the formation of a large amount of defects[20]. No symmetry-forbidden peaks at low angle (2θ at ca. 4 and 6°) were observed in the PXRD pattern of the sample, suggesting the absence of missing cluster defects that would lead to an ordered structure with reo topology (Supplementary Fig. 2)[22]. Thus, we hypothesized the presence of missing linkers in our material. Fourier transform infrared spectroscopy (FTIR) demonstrated the absence of uncoordinated carboxylic acid residual groups in the washed materials, in agreement with the presence of linker defects rather than missing oxoclusters (Supplementary Fig. 5). Thermogravimetric analysis (TGA) under oxygen atmosphere evidenced that the 40 nm UiO-66 nanoMOF exhibited a very low ligand-to-metal ratio (linker: $Zr_6$ = 3.96:1, Supplementary Fig. 6), in agreement with a high missing linker content. The number of missing ligands in our sample corresponds to one of the most defective UiO-66 reported so far and concomitantly, to the best of our knowledge, the surface area of 40 nm UiO-66 represents a comparably high value compared to the state-of-the-art[23,34,35,40].

The general method we developed to control the particle size is illustrated in Fig. 2a. As our first attempt led to highly defective 40 nm UiO-66 while using only 10 mL EtOH with 50 mg of BDC, the reaction media was further diluted stepwise with EtOH (20, 40, 80 mL, respectively) enabling a better dissolution of the 1, 4 BDC ligand. Interestingly, this led to a dramatic reduction of nanoparticle size, as evidenced by the PXRD patterns in Fig. 2b. Notably, when the volume of ethanol reached 80 mL, only a broad envelop of the main characteristic diffraction peaks of UiO-66 could be observed due to the considerable loss of long-range order, which is consistent with the Scherrer equation. The TEM images (Supplementary Figs. 3, 7, 8, and Figs. 2c, d) evidenced the precise control of downsizing down to between 4 nm and 6 nm. Nevertheless, the crystal lattice planes can still be observed by TEM (Fig. 2d), confirming the quality of the nanoparticles. The selected area electron diffraction (SAED) pattern only showed the characteristic rings of the UiO-66 nanoparticles with different crystal orientations (Fig. 2e), further proving the quality of these nanoMOFs, as well as homogeneous size distribution. Further enlargement of Fig. 2d(i) clearly shows the pores and lattice of a 4.4 nm nanoparticle, in good accordance with the structural model from Fig. 2d(ii). From the profile analysis on HRTEM images along (220) and (011) directions (Fig. 2f), the distances between two adjacent $Zr_6$ oxoclusters are highly homogeneous with an average value of 1.1 nm that is close to the theoretical one (1.2 nm), suggesting the absence of oxocluster defects in our ultrasmall UiO-66 (HD-US-UiO-66). To be noted, such a small particle corresponds to only ca. 2 unit-cell dimensions, i.e., 8 unit-cells or 12 octahedral-cages per nanoparticle. This system therefore lies at the frontier between nanocrystals and discrete metal-organic polyhedra. However, in contrast with our ultrasmall particles (see after for the description of the thermal and chemical stability), Metal−Organic Cages/Polyhedra (MOCs/MOPs) often suffer strongly from poor chemical/hydrolytic stabilities and structural collapse upon activation, preventing their applications[41].

According to TGA, all the different as-prepared nanoparticles exhibited very similar linker content, with close to 2 missing linkers per formula (Fig. 2g). This, once combined with liquid phase nuclear

magnetic resonance (NMR) analysis (Supplementary Fig. 9) and Energy Dispersive X-ray spectroscopy (EDX) (Supplementary Fig. 10), leads to a general formula of $Zr_6O_4(OH)_4(BDC)_{3.9}(C_2H_3O_2)_{0.8}(H_2O)_{2.9}Cl_{0.5}$. Note that the connection of crystal size to defectiveness in our syntheses is, to the best our knowledge, the first of its kind and allows for achieving <5 nm MOF nanoparticles with defectiveness of up to 4.2 missing linkers per oxocluster, by far exceeding the commonly reported values (Supplementary Fig. 11)[18]. Nitrogen porosimetry at 77 K on the activated HD-US-UiO-66 particles evidenced in all cases a high sorption capacity (from 404 cm$^3$/g to 260 cm$^3$/g, Fig. 2h) associated with a hysteresis. A decrease in $N_2$ adsorption capacity occurred upon downsizing, in line with a progressive increase of the external to internal surface ratio. Pore size distribution analyses (DFT model) indicated overall a preserved pore size (Fig. 2i), which is once again in line with the constant missing linker content. For the sake of comparison, we followed the conventional MIDA, as well as the synthetic parameters of 40 nm UiO-66 and carried out a set of synthetic experiments by reducing the amount of acetic acid. Noteworthy, the preparation of smaller particles, down to 5 nm, denoted as MI-US-UiO-66 (modulator-induced ultrasmall UiO-66), is associated with lower number of defects (2.6 missing linkers per $Zr_6$ oxoclusters) in good accordance with the previous findings where the missing linker defect content strongly depended on the modulator quantity (see SI for details).

To further investigate the nature of the missing linker defects at atomic level, in situ FTIR spectroscopy in presence of acetonitrile-$d_3$ (CD$_3$CN) vapors was performed. The acidity of pristine UiO-66(Zr) is mostly assigned to its intrinsic Brønsted acid sites (four μ$_3$-OH), while, upon high temperature activation, additional Lewis acid sites associated to the defects (missing linker) are present. When introducing CD$_3$CN aliquots to 10 torr equilibrium pressure, three vibrational bands could be observed at 2306, 2301, and 2276 cm$^{-1}$, associated with the chemisorption of CD$_3$CN on different Lewis and Brønsted acid sites (Fig. 3). The ν(CN) bands at 2306 and 2301 cm$^{-1}$ mostly dominate the spectra in the first three doses, indicating a strong interaction between the Lewis acid sites (Zr$^{4+}$) of the MOF and CD$_3$CN. Interestingly, these Zr-CD$_3$CN bands showed a slight blue shift, from 2296 cm$^{-1}$ to 2306 and 2301 cm$^{-1}$, in comparison to the constant peak position of the physisorbed CD$_3$CN (2261 cm$^{-1}$) in other reported works[42,43]. This clearly indicates a higher acidic strength of the sites, likely promoted by the large concentration of defects. Their concentration could be calculated by integrating the corresponding bands vs. the molar amount of CD$_3$CN introduced. The obtained value of 1.19 mmol/g corresponds to a much larger number of Lewis acid sites than commonly reported for much larger-sized defective UiO-66 via MIDA, e.g., typically around 300 μmol/g[42], which further demonstrates that our ultra-small UiO-66 nanoparticles exhibit a much higher degree of defects, being therefore particularly interesting for Lewis acid based catalysis.

Several hypotheses can be proposed to understand the formation of HD-US-UiO-66. First, the pre-formed $Zr_6$ acetate oxoclusters establish the pH of the solution near 4, which favors ligand deprotonation of the carboxylic groups of the ligand and thus leads to a faster nucleation upon substitution of the terminal acetates from the oxoclusters in the presence of the dicarboxylate moieties. Then, the dilution upon addition of ethanol impairs the effective collision rate, and once combined with the high modulator content, might limit the crystal growth as well as Ostwald ripening. However, upon downsizing, the reaction kinetics becomes faster, from 3 h for the largest particles to less than 1 h for the 4−6 nm particles. Thus, other parameters are likely in play. For instance, the solubility of the ligand is limited at RT in ethanol and therefore, the proportion of ligand that is solubilized increases with the ethanol dilution, which favors a faster kinetics for the smallest particles. To validate this hypothesis, considering the much better solubility of BDC in DMF, we replaced 50% of EtOH by

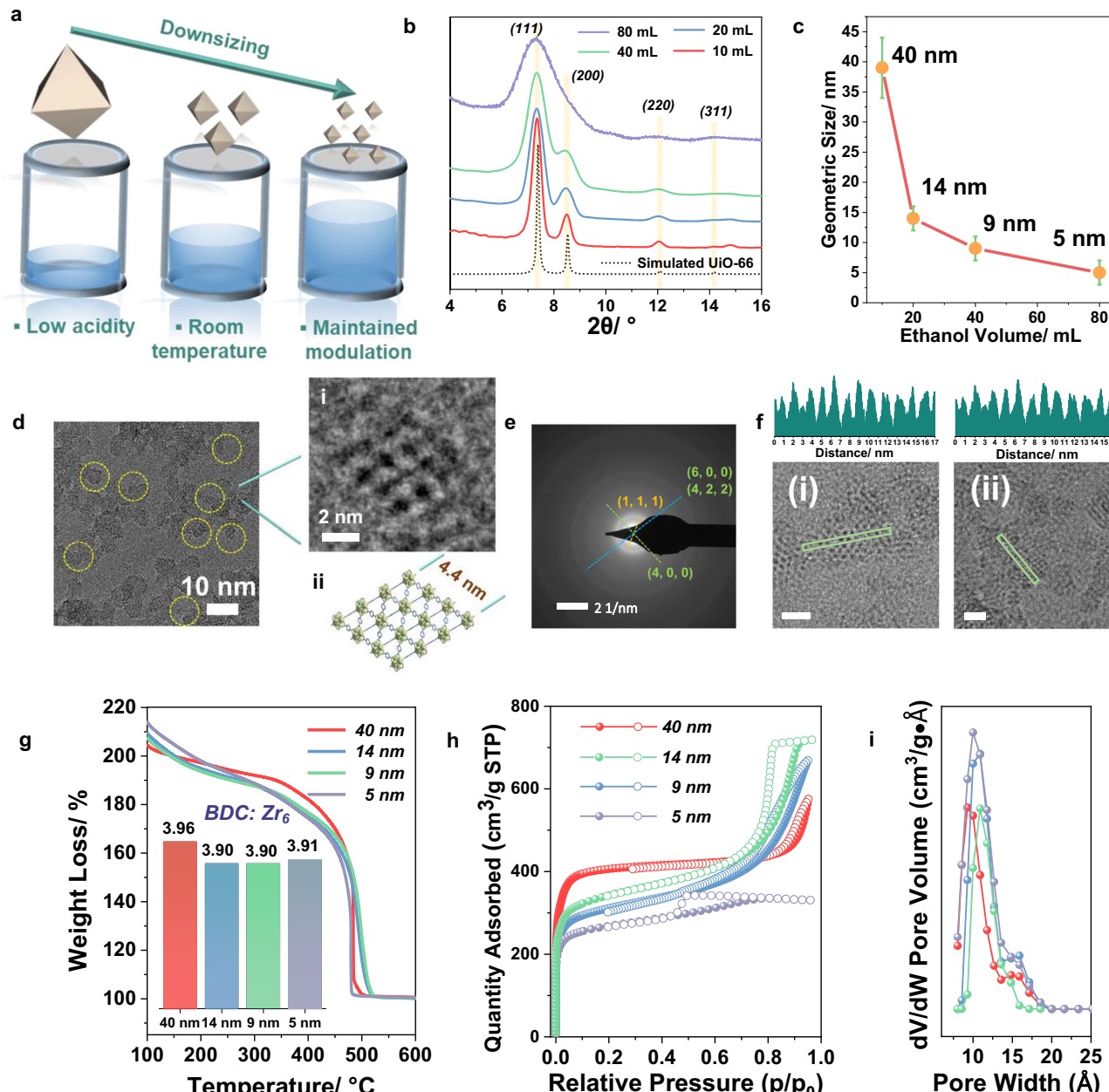

**Fig. 2 | Characterizations of the synthesized UiO-66 nanoparticles. a** Schematic diagram of our strategy, (**b**) powder X-ray diffraction (PXRD) ($\lambda_{Cu}$ = 1.5406 Å) patterns of UiO-66 synthesized with different volumes of EtOH, the yellow lines evidence the diffraction peaks from the calculated pattern, (**c**) statistical mean size of the synthesized UiO-66, (**d**) TEM image of the 5 nm UiO-66 (obtained with 80 mL EtOH, highlighted using yellow circles), i) enlarged selected zone, ii) structure of UiO-66 viewed from (101) axis direction, (**e**) SAED pattern of the 5 nm UiO-66, (**f**) High-resolution Transmission Electron Microscope (HRTEM) images of HD-US-UiO-

66 and their contrast intensity profiles, viewed along (i) (220) and (ii) (011) directions, scale bar = 5 nm, the green rectangles represent the selected regions for contrast analysis, (**g**) TGA under oxygen atmosphere (scan rate of 3 °C/min) of the UiO-66 with different sizes, inserted bar chart: linker to $Zr_6$ ratio of the different samples; (**h**) 77 K $N_2$ sorption isotherms of UiO-66 with different sizes (P/$P_0$ = 1 bar), adsorption and desorption are represented by filled spheres, and open spheres, respectively, i) pore size distribution for different sizes of UiO-66 (same color label as in (**h**)).

DMF whilst keeping all other synthetic parameters constant. The resulting nanoparticles were found to be only ca. 8 nm instead of 40 nm in comparison with the use of EtOH (Supplementary Fig. 18), which corroborates the influence of the linker's solubility on the kinetics. Such a bottleneck was observed previously by some of us when increasing the size of the dicarboxylic acid organic spacer over the crystallization under solvothermal conditions of UiO-66(Zr) and its extended analogs[44].

To gain further understanding about the formation mechanism of the HD-US-UiO-66, in situ time-dependent dynamic light scattering experiments (TD-DLS) were conducted. Figure 4a showed the

evolution of particle size as a function of crystallization time of HD-US-UiO-66. The fast size increase process in the first 15 min was attributed to the formation of MOF nuclei. The MOF's growth was observed in the range between 15 min and 130 min and reached saturation after 130 min with a hydrodynamic size at ca. 18 nm. The clear slope revealed that the crystallization of HD-US-UiO-66 follows a growth-dominated process, in good agreement with the role of linker dissolution. The very low polydispersity index (Pdi) (Supplementary Fig. 19) implied a homogeneous nucleation followed by crystal growth of UiO-66 in solution. To confirm the growth kinetics of HD-US-UiO-66, an ex-situ HRTEM/STEM study was carried out after 0, 1, 120, and

180 min. Figure 4b shows that only $Zr_6$ oxoclusters (ca. 0.6 nm) were observed before the introduction of BDC. Ultra-small nanoparticles (around 1.3 nm) were formed very quickly as soon as the ligand was added (1 min), which indicates the very fast nucleation of UiO-66 nanoparticles in solution, in agreement with the stage (i) shown in Fig. 4a. The TEM images at 120 and 180 min demonstrated the growth of the small MOF nuclei and the saturation of nanoparticles growth. These results are fully consistent with TD-DLS, confirming that the ligand dissolution acts as the bottleneck in controlling the MOF growth. Notably, the in situ TD-DLS not only sheds light on the crystallization process but also strongly highlighted the excellent colloidal stability of the HD-US-UiO-66. This stability, correlated to the highly positive charge evaluated by Zeta potential analysis, was observed whatever the nanocrystal size (Figs. S20, S21) and is a strong asset for their solution processability in a view of applications such as ultrathin film fabrication, drug delivery, sensing and electronics, among others[45].

One very appealing feature of MOFs is their ligand functionalization to achieve desired properties. Thus, we extended the synthetic strategy to produce other functional HD-US-UiO-66-X derivatives,

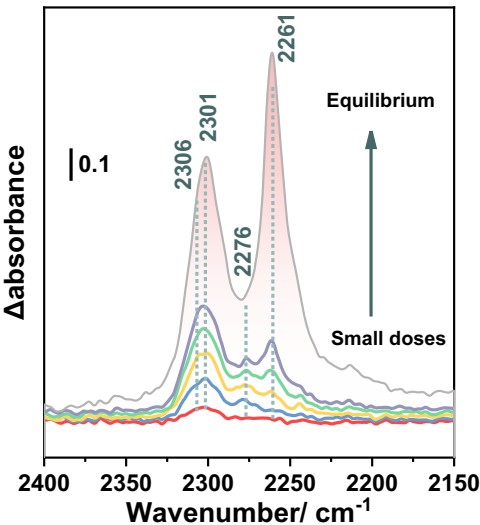

**Fig. 3 | Characterization of the accessible acid sites by in situ FTIR.** In situ FTIR spectra at 298 K of $CD_3CN$ (red to gray, probe small doses to up to 10 torr equilibrium pressure) adsorbed on HD-US-UiO-66 (5 nm), the peaks position and peak area are shown in dotted lines and red area, respectively, the small doses and equilibrium indicate the increasing $CD_3CN$ dosing from 0.2 torr to saturated 6.4 torr.

including X = $NH_2$, $NO_2$, $(OH)_2$, and Br, as well as replacing $Zr_6$ by $Hf_6$ oxoclusters and BDC-based planar linkers by fumarate. All these ultra-small nanoMOFs exhibited 2-3 unit-cell particle sizes and very similar connectivities (ca. 3.3–4 linkers per formula) as the pristine HD-US-UiO-66 (see detailed analysis in SI).

### Heterogeneous catalysis

In light of the properties of our HD-US-UiO-66-X, we decided to explore their use in heterogeneous catalysis[6,18,46]. The challenging hydrolysis of the peptide bond in glycylglycine (GG) was selected as the model reaction to investigate the significance of downsizing/ defect formation of MOF nanoparticles on the overall catalytic performance (Fig. 5a). HD-US-UiO-66-$NH_2$ nanoMOF was subsequently selected for catalysis due to the potential H-bonding that could occur between the peptide substrate and the -$NH_2$ group on the ligand, or between the -$NH_2$ group and $H_2O$ nucleophile, which could contribute to the overall catalysis. As anticipated, the use of HD-US-UiO-66-$NH_2$ led to ca. 3 times higher reactivity compared to conventional UiO-66-$NH_2$ particles (ca. 200 nm) reported previously (Fig. 5b)[47]. Such an enhancement might be due to the significantly expanded external surface area and/or the large defect content. Thus, we have compared the performance of HD-US-UiO-66-$NH_2$ and MI-US-UiO-66-$NH_2$ nanoMOFs with the previous study. Noteworthy, although MI-US-UiO-66-$NH_2$ showed better reactivity compared to the previous study, it performed worse (ca. 2 times) than HD-US-UiO-66-$NH_2$, despite their very similar particle size. This suggests that maximizing the amount of missing linker defects is critical to enhance the reactivity of UiO-66 derivatives towards peptide bond hydrolysis, in combination with decreasing the particle size. Notably, the reactivity here is comparable to benchmarks materials such as the "superactive" MOF-808 (35 nm), which exhibited a GG hydrolysis rate of $2.69 \times 10^{-4}\,s^{-1}$.[48]

The recyclability of HD-US-UiO-66-$NH_2$ nanoMOF was tested over five subsequent reaction cycles (Supplementary Fig. 35), with the slight loss of reactivity being only recorded after the 4th cycle, where the reactivity dropped to 80% of the performance observed in cycle 1. This is however still an excellent recyclability, especially when compared to benchmark materials, such as MIP-201 and MOF-808[48,49], which suffered from lower recyclability (only 45% and 20% activity after 5 cycles, respectively, Fig. 5d), and highlights the water stability of these ultra-small highly defective nanoMOFs.

The specificity of the catalyst was further studied, as GG may be hydrolyzed to G, or may undergo cyclisation (amide bond formation), forming cyclic GG (cGG) (Supplementary Fig. 32). By using HD-UiO-66-$NH_2$ nanoMOFs of two different sizes (4 nm vs 200 nm), the influence on particle size and external vs internal surface area on the reaction specificity was examined. The 200 nm UiO-66-$NH_2$ is labeled as HD-200-UiO-66-$NH_2$, and was fully characterized prior to catalytic reaction

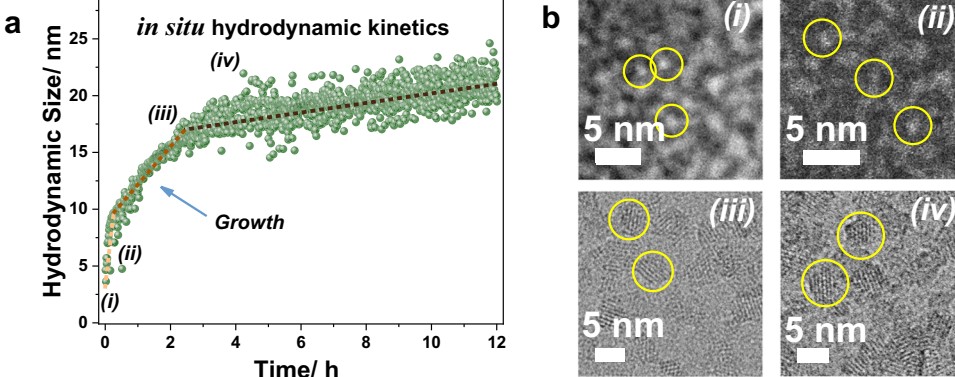

**Fig. 4 | In situ and ex-situ characterizations of the MOF growth. a** Hydrodynamic size of HD-US-UiO-66 colloids ($T = 25\,°C$) determined by in situ time-dependent DLS (time resolution $t_R = 120\,s$), and the (**b**) ex-situ HAADF-STEM (0 = i, 1 min = ii) and HRTEM (120 min = iii, 180 min = iv) images of HD-US-UiO-66 at different times.

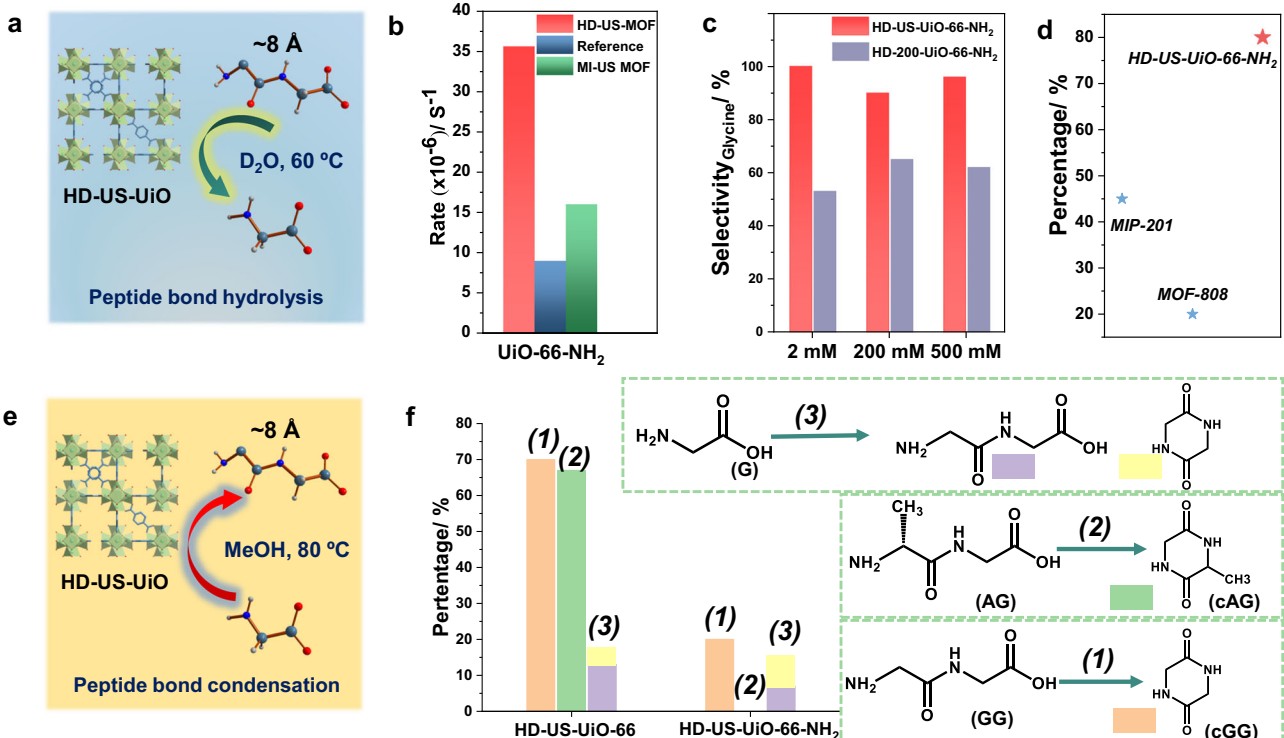

**Fig. 5 | Catalytic performance evaluation of the HD-US-UiO-66-X. a** Illustration of peptide hydrolysis using HD-US-UiO-66, (**b**) Pseudo first order hydrolysis rate of glycylglycine (GG) to glycine (G) using HD-US-UiO-66-NH$_2$ and MI-US-UiO-66-NH$_2$ nanoMOFs, reference refers to the value reported in the previous studies under the same conditions[47], (**c**) Selectivity of hydrolysis by HD-US-UiO-66-NH$_2$ and HD-200-UiO-66-NH$_2$ in producing the desired product G, as the starting concentration of GG increases from 2 mM to 500 mM, (**d**) Recyclability of HD-US-UiO-66-NH$_2$ over 5 reaction cycles in comparison to best-performing MOFs to date, percentage compared to yield of cycle 1, (**e**) illustration of amide bond condensation using HD-US-UiO-66, in MeOH, (**f**) Amide bond formation yield with HD-US-UiO-66 and HD-US-UiO-66-NH$_2$ starting from G, GG, and AG.

(Supplementary Figs. 36–38). Interestingly, the HD-US-UiO-66-NH$_2$ was found to be more selective, consistently giving a greater proportion of hydrolysis product G, rather than cGG, regardless of the starting concentration of GG substrate (Fig. 5c). The HD-200-UiO-66-NH$_2$ produced considerably more cGG (40–50% of product), likely due to the increased internal surface area compared to HD-US-UiO-66-NH$_2$, which promotes amide bond condensation in the absence of water in the more hydrophobic pores, rather than hydrolysis reaction at the water exposed external surface of the nanoMOF.

As cGG was found to be a side product of hydrolysis with HD-200-UiO-66-NH$_2$, the nanoMOFs were further examined for their ability to catalyze the peptide bond formation between amino acids and small peptides. The condensation reaction can be promoted instead of hydrolysis simply by changing the solvent from water to methanol[50]. Both HD-US-UiO-66-NH$_2$ and HD-US-UiO-66 nanoMOFs showed the ability to promote amide bond formation in MeOH (Fig. 5e, f), but the HD-US-UiO-66 exhibited much higher reactivity than the functionalized HD-US-UiO-66-NH$_2$. Similar to the previous report[49], for both MOFs the intramolecular peptide bond formation was favored over intermolecular condensation between individual glycine molecules (Reaction 3, Fig. 5f). Interestingly, the functionalized HD-US-UiO-66-NH$_2$ was ineffective in forming cGG when starting from GG (Reaction 2), but the fact that cGG was observed in other two reactions shown in Fig. 5f, suggests that the efficiency of intramolecular bond formation is influenced by substrate interaction with the MOF, rather than by a reduced catalytic activity. These differences in the interactions could be influenced by the presence of the electron donating -NH$_2$ groups on the MOF, as well as by the MOF pore sizes, as the functionalized MOF exhibits slightly smaller pores due to the steric hindrance of the -NH$_2$ group, which increases the diffusion barrier of the substrate into and out of the pores of the MOF.

In this work, we have reported a general strategy for the preparation of a series of highly defective (35–45% missing linker) and ultra-small (4–6 nm) UiO-66-based nanocrystals under fully sustainable conditions, suggesting feasibility towards upscaling. Crystal growth acts as a bottleneck in crystallization, as evidenced by in situ TD-DLS with ex-situ HRTEM/STEM, and can be manipulated by simply using additional solvent. Missing linker defects have been assessed by multiple advanced techniques, including PXRD, FTIR, TGA, HRTEM and in situ FTIR spectroscopy coupled with CD$_3$CN probe, which revealed the importance of Lewis-acidity of the HD-US-UiO-66. The resulting HD-US-UiO-66-X showed excellent catalytic performance in both peptide bond hydrolysis and formation with catalytic reactivity, selectivity, product recovery efficiency, and recyclability compared to other reported materials. Detailed investigation of the influence of defects, particle size and functionalization on the catalytic activity of the nanoMOFs provided unique insights into the key parameters that influence the reactivity, and as such demonstrate how nanoMOFs can be tuned to show specific and selective reactivity through precise control of catalysts' properties. Therefore, the discoveries reported here might further promote the development of nanoMOFs as heterogeneous catalysts having dual functions and performance enhancements in varying aspects. Furthermore, the materials presented here may be also used for the development of sensing/optical devices (preliminary results in Supplementary Discussion, and Supplementary Fig. 24), membranes, nanomedicine formulations and to explore other fundamental size-dependent properties.

## Methods

### Synthesis of Zr$_6$ oxoclusters

ZrCl$_4$ (2 g, 8.4 mmol) was added into a mixture of 3 mL of glacial acetic acid and 5 mL of isopropanol under stirring at 500 rpm and heated at

120 °C for 60 min. The product was collected either through suction filtration or centrifugation at 10,000 rpm. The collected white solid was subsequently washed with acetone twice and dried under vacuum at RT. The synthesized $Zr_6$ oxoclusters could be stored in ambient condition for at least 12 months without properties change. The prepared $Zr_6$ oxoclusters have the following formula: $Zr_6O_4(OH)_4(C_2H_3O_2)_8(H_2O)_2Cl_3$.

## Synthesis of Hf₆ oxoclusters

$HfCl_4$ (0.7 g, 2.2 mmol) was added into a mixture of 4 mL of glacial acetic acid and 18 mL of isopropanol under stirring at 500 rpm and heated at 100 °C for 60 min. The product was collected either through suction filtration or centrifugation at 10,000 rpm. The collected white solid was subsequently washed with acetone twice and dried under vacuum at RT.

## Synthesis of 40 nm UiO-66

$Zr_6$ oxoclusters (0.06 mmol, 75 mg) were dispersed in acetic acid (0.5 mL, 8.75 mmol) under stirring at 600 rpm. $H_2O$ (1.25 mL) was subsequently added, and the reaction mixture was stirred until it became completely clear. 10 mL of ethanol was introduced into the solution followed by the immediate addition of 1,4-benzenedicarboxylic acid (BDC, 50 mg, 0.3 mmol), and the reaction was stirred for 3 h at room temperature. The resulting solution was centrifuged at 14,500 rpm for 45 min and then washed twice with ethanol (14,500 rpm, 45 min). The collected solid was dried under vacuum for 3 h for characterizations and applications.

## Synthesis of MI-US-UiO-66 (5 nm)

$Zr_6$ oxoclusters (0.06 mmol, 75 mg) were dispersed in acetic acid (0.025 mL, 0.44 mmol) under stirring at 600 rpm. $H_2O$ (1.25 mL) was subsequently added, and the reaction mixture was stirred until it became completely clear. 10 mL of ethanol was introduced into the solution followed by the immediate addition of 1,4-benzenedicarboxylic acid (BDC, 50 mg, 0.3 mmol), and the reaction was stirred for 1 h at room temperature. The resulting solution was centrifuged at 14,500 rpm for 45 min and then washed twice with ethanol (14,500 rpm, 45 min). The collected solid was dried under vacuum for 3 h for characterizations and applications. Modulator-induced size control can be applied with different amount of acetic acid used in the synthesis batch.

## Synthesis of HD-US-UiO-66 (5 nm)

$Zr_6$ oxoclusters (0.06 mmol, 75 mg) were dispersed in acetic acid (0.5 mL, 8.75 mmol) under stirring at 600 rpm. $H_2O$ (1.25 mL) was subsequently added, and the reaction mixture was stirred until it became completely clear. 80 mL of ethanol was introduced into the solution followed by the immediate addition of 1,4-benzenedicarboxylic acid (BDC, 50 mg, 0.3 mmol), and the reaction was stirred for 2 h at room temperature. The resulting solution was evaporated by rotary evaporation at room temperature until ~10 mL volume was left. The colloidal suspension was centrifuged at 14,500 rpm for 60 min and then washed twice with the mixture of 20 mL of acetone and 20 mL of ethanol (14,500 rpm, 1.5 h). The collected solid was dried under vacuum for 3 h for characterizations and applications. The size control approach here can be applied with different amount of ethanol used in the synthesis batch.

## Synthesis of HD-US-UiO-66-NH₂ (4 nm)

$Zr_6$ oxoclusters (0.06 mmol, 75 mg) were dispersed in acetic acid (0.5 mL, 8.75 mmol) under stirring at 600 rpm. $H_2O$ (1.25 mL) was subsequently added, and the reaction mixture was stirred until it became completely clear. 80 mL of ethanol was introduced into the solution followed by the immediate addition of 2-aminobenzene−1,4-dicarboxylic acid (BDC-$NH_2$, 55 mg, 0.3 mmol), and the reaction was

stirred for 1 h at room temperature. The resulting solution was evaporated by rotary evaporation at room temperature until ~10 mL volume was left. The colloidal suspension was centrifuged at 14,500 rpm for 60 min and then washed twice with the mixture of 20 mL of acetone and 20 mL of ethanol (14,500 rpm, 1.5 h). The collected solid was dried under vacuum for 3 h for characterizations and applications.

## Glycylglycine hydrolysis

Prior to hydrolysis, MOFs were activated at 120 °C for 20 h. To a 1 mL glass vial was added 2 μmoles of MOF and 950 μl $D_2O$. Next, 50 μl of a 40 mM solution of Glycylglycine in $D_2O$ was added, the mixture pD was adjusted to 7.4 and incubated at 60 °C with stirring. Vessel was prepared per time point in triplicate. Reactions were stopped at 1 h, 2 h, 5 h, 20 h, 24 h, and 48 h and centrifuged at 14,000 rpm for 10 minutes. 500 μl of supernatant was analyzed with ¹H-NMR using 3 μl of 0.1 M TMSP$_{d4}$ internal standard. MOFs were washed in acetone and analyzed with PXRD after reaction to check their structure integrity.

For recycling experiments, after reaction, the MOFs were washed in $D_2O$ overnight to remove adsorbed substrate and product, followed by washing in 10 mL of acetone and drying in the oven for 8 h at 110 °C. After which the reaction was repeated.

## Peptide bond formation

MOFs were used after activation. To a 10 mL crimp cap vial was added 5 μmoles of MOF and 50 μmoles of substrate (glycylglycine, L-alanylglycine or glycine) followed by 1 mL methanol. The mixtures were sealed and stirred, and then incubated at 80 °C for 24 h with stirring. After incubation, samples were diluted with $D_2O$ (1 mL) and stirred at room temperature for 1 h to elute substrates and products from the MOFs. After which the supernatant was collected via centrifugation at 14,000 rpm for 10 min. 100 μL of supernatant was then diluted with $D_2O$ to give a final volume of 500 μL, which was analyzed with ¹H-NMR using 3 μl of 0.1 M TMSP-$d_4$ internal standard.

## Data availability

All data supporting the finding of this study are available from the corresponding authors upon request. Source data are provided with this paper.

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

## Acknowledgements

S.D., C. Serre, and M.D. acknowledge the European Union's Horizon 2020 research and innovation program under grant agreement No. 831975 (MOF4AIR project) for providing financial support. C. Simms thanks the Research Foundation-Flanders (FWO) for the fellowship grant (68090/11C9320N). The authors appreciate the help of Dr. X. Xu for TEM measurements. S.D., A.T., G.P., and C. Serre acknowledged the Paris

region, through the DIM Respore priority project, for the access to HRTEM.

## Author contributions

S.D., A.T., and C. Serre conceived the research. S.D., C. Simms, and T.V. designed the experiments and analyzed the data. G.P. conducted the STEM/HRTEM analysis. M.D. supervised the in situ FTIR investigations. S.D. wrote the original draft with the help of C. Simms. All authors contributed in the paper discussion and revision. A.T. and C. Serre supervised the project.

## Competing interests

The authors declare no competing interests.
