## [Peer Review File · Nature Communications]

Highly defective ultra-small tetravalent MOF nanocrystalsREVIEWER COMMENTS

Reviewer #1 (Remarks to the Author):

Dai, Tissot, Parac-Vogt, Serre and co-workers report the methodology for the synthesis of Zr and Hf containing MOF nanocrystals with a high percentage of defects. The work was mainly performed on the UiO-66 Zr(IV) system (and MOF-801) where the generation of defects have previously been well characterised with increasing concentrations of modulators (monocarboxylic acids) used. The generation of MOF nanocrystals with a high percentage of defects is of importance for catalysis where higher reactivity can be achieved. The research is highly topical, relevant to numerous applications and an important contribution to the MOF literature. The work will be of interest to the broad readership of Nature Communications. I thoroughly enjoyed reading the manuscript - all experiments were performed to a high level and conclusions made from experimental work are sound. The mechanism of MOF nanocrystal formation and the role of the solvent were well considered.

I only had a minor comment regarding the yields obtained from the synthesis of the MOF nanocrystals - could these please be included in the experimental section? Does the yield change the size of particle obtained or with the substituent on the terephthalate ligand used for the synthesis of UiO-66?

The synthesis of the MOF nanocrystals are performed at a small scale (75 mg of the starting Zr cluster) - is it possible to scale up the synthesis?

Reviewer #2 (Remarks to the Author):

Dear Authors and Editor

This manuscript reports a novel synthetic approach for the production of Zr- or Hf-based UiO-66 nanoparticles. The method proposed by the Authors allows for a useful control over the size of the produced particles down to a remarkably small length of less than 10 nanometers, which is undoubtedly of great interest for MOF applications traditionally hampered by mass transport limitations (e.g. heterogeneous catalysis). The study also shows some synthetic influence over the type and amount of defects, although their characterization—notoriously challenging—lacks the level of details and accuracy for substantiating some of the Authors' claims. Last but not least, the proposed synthetic strategy is more sustainable and environmentally friendly than most MOF syntheses, which still largely rely on the use of N,N-dimethylformamide and lengthy solvothermal conditions. After describing a reasonably thorough characterization of the products and providing some useful—although mostly speculative—considerations on their formation, the Authors conclude by testing the versatility of their method on differently functionalized forms of UiO-66, and by assessing the catalytic performance of some of them compared to forms of the same materials having different crystallinity.

Overall, I consider this work of high interest for the community and I am positive its quality and novelty qualify it for publication in Nature Communications. However, I believe some parts of this study lack clarity, information, or contain possibly incorrect statements. Therefore, a revision is essential to make it suitable for acceptance and publication. My concerns and suggestions are reported as follows.

1. Scientific concerns

1.1. At the beginning of the "Results" section, the authors classify the ultra-small and highly defective UiO-66 particles as "highly crystalline". This is quite confusing, since—in simple terms—crystallinity is the degree of translational order in a material (direct space definition) or the 'sharpness' of peaks in their diffraction pattern (reciprocal space definition; <https://dictionary.iucr.org/Crystal>). The confusing aspect is that this entire study focuses on achieving the lowest possible crystallinity (smallest crystal size, highest amount of defects). The tendency of referring to these products as 'highly crystalline' is, in my view, part of a general phenomenon leading reticular chemists to claim their compounds are 'highly crystalline' as if this

was a must-have virtue for a material to be useful. Needless to say that this is not based on reality, and the properties of highly defective and ultra-small UiO-66 particles prove further that low crystallinity is sometimes extremely interesting. It is, as a matter of fact, what the Authors pursue in their work. I would therefore encourage them to revise their manuscript by re-thinking the best phrasing of all sentences where the concept of 'crystallinity' or 'crystalline' is used, since this term is so often abused and misused in the MOF&COF community.

1.2. Still in the very first line of the "Results" section, the authors invoke the concept of "maximal defect content". Is it certain that the maximal content of defects is achieved, and particles with more defects are theoretically impossible to form? If so, the authors should demonstrate this point since, in my view, it is a rather strong and possibly inaccurate claim.

1.3. When the Authors describe that the particle size value of 44 nm calculated by the Scherrer equation is very close to that observed by TEM, they conclude that the particles are well-crystalline. I do not see the connection here. The most obvious conclusion that one could draw is that, if we consider the Scherrer equation method reliable, the sample observed by TEM has been fairly representative of the bulk. The fact that the Scherrer equation produces results matching the observation by TEM is not a sign that the particles are well-crystalline, especially since the Scherrer equation itself shows that, as desired, the crystallinity is relatively low: a few tens of unit cells down to less than ten.

1.4. The Authors mention that the "absence of uncoordinated carboxylic acid residual groups in the washed materials" is "in agreement with the presence of linker defects". I cannot see the link between the two aspects. UiO-66 particles that are free of uncoordinated carboxylic groups can still be fully-connected structures without any defect. I believe that either the authors created an incorrect correlation between IR-based evidence and presence of linker defects, or I did not correctly understand their statement. In the latter case, I kindly ask the authors to make this sentence clearer.

1.5. When discussing the TEM imaging results of the smallest particles, the Authors correlate being able to resolve all clusters in the images to having missing linkers defects. I don't see how this link is sensible. From the available TEM images there is no direct information on the presence or absence of linkers and it is therefore not possible to make any conclusion concerning linker defects. Moreover, especially with respect to these small particles, deviations in cluster-to-linker ratios are not necessarily indicative of missing linkers in the framework internal structure. Indeed, if we consider a 2x2x2 unit cells UiO-66 particle and hypothesize that modulators acted as capping agents to obtain a fully cluster-terminated particle, its composition would be 63 clusters and 240 linkers when all linkers are present. This would mean a linker-Zr6 ratio of ca. 3.809 even without defects within the structure. Since the ratio that leads the Authors to claim that linkers are missing is actually higher (ca. 3.9), it could mean that these hypothetical 2x2x2 particles, although having no defects, have an excess of linker—which can only be attached to their outside. Of course, these considerations become less relevant for larger particles, where the evidence for missing linker defects is stronger since the surface termination plays a weaker role in defining the linker-cluster ratio. Nevertheless, I encourage the Authors to invest some efforts in commenting these aspects, thereby adding some healthy caution that I am sure would be highly appreciated by most readers.

2. Missing information

2.1. In the introduction, the Authors discuss the advantages of downsizing MOFs, providing useful examples of "enhanced properties (i.e., catalysis, sensing, biomedicine,...)". Right afterwards, the authors also mention "novel features", although it is not clear what these features are, and no reference is provided. It would be useful to mention these novel features between brackets and provide appropriate references to make this addition meaningful.

2.2. Later in the introduction, the Authors report how the MIDA approach has been mainly used on "UiO-66 with missing linker defect content of ca. 10-20%." This statement requires at least one reference.

2.3. Missing clusters defects were deemed absent based on XRD profiles, as “no symmetry-forbidden peak at low-angle” were observed. The authors mention that these peaks should be visible below $2\theta = 7^\circ$, but it would be most informative to mention at which 2θ values these extra peaks are expected, since this is a piece of information that can be easily provided.

2.4. In Figure 2, the scale bar of the diffraction pattern is reported in pixel instead of meaningful units such as $1/\text{nm}$ or $1/\text{\AA}$. Please insert a correct scale bar for this picture.

2.5. Still in Figure 2, in the stacked diffraction pattern the label of the last reflection appears to be wrong. Not only the (0 2 2) reflection is already reported to its left, but UiO-66 has two different reflections in this region: (1 1 3) and (2 2 2). Which one did the Authors intend to highlight?

2.6. I understood that the linker, once added in the syntheses of the nanoparticles, is not fully dissolved but instead forms a suspension. I think this is very important to state more explicitly, since this would mean that the growth of UiO-66 nanoparticles might proceed via heterogeneous nucleation on the undissolved linker particulate, and also that linker solubility strongly influences the kinetics of the particles' growth.

2.7. The Authors mention something interesting: “although Metal–Organic Cages/Polyhedra (MOCs/ MOPs) have intrinsically discrete supramolecular architectures, they often suffer strongly from poor chemical/hydrolytic stabilities and structural collapse upon activation”. Once having read this, I immediately wondered how this transfers to such small $2 \times 2 \times 2$ unit cell particles or analogous size. This appears to be implied by the final sentence, but the point is not clarified further, leaving the readers wondering: is this true also for these ultra small particles when their size is extremely small? To which extent? Does the fact that this is not strictly a cage, but a slightly larger coordination assembly, change these considerations? And does it change them only quantitatively or also qualitatively? I believe that, for the sake of being informative and transparent, the Authors should briefly address these aspects when concluding this paragraph.

2.8. When rationalizing the formation of HD-US-UiO-66, the Authors mention a pH condition of ca. 4. It is not clear where this value originates from, especially since it is not mentioned in the synthetic procedures (SI). Probably, this value refers to a previous study, in which case the Authors are kindly advised to add a dedicated reference.

2.9. The Authors mention an “excellent colloidal stability” assessed by in situ TD-DLS. Is there a most-likely origin of this stability? If a sound hypothesis could be formulated, it would be interesting if the Authors could briefly introduce it after the stability assessment.

2.10. There is some useful information missing in the experimental procedure reported in the SI, as follows.

- In the preparation of the Zr₆ oxoclusters it is not clear how the chemical composition of the clusters has been determined. If this refers to a past study, it should be stated explicitly.
- Essentially every synthesis procedure mentions a “completely colorless” reaction mixture. Since I assume the reactants are colorless to begin with, I guess the Authors meant “clear” and not “colorless”.
- The appearance of the solution after linker addition is not followed by whether this resulted in a clear solution or in a suspension. From what I understood, poor linker solubility often resulted in suspensions being formed. This should be mentioned explicitly since it is a critical piece of information to better understand the conditions under which the products are formed.
- Several synthesis procedures report a volume of acetic acid without clarifying its concentration. Please add this piece of information.

2.11. In the SI Figure S26, the images show some ring patterns that are extremely difficult to distinguish from the background. I advise the Authors to change intensity scale and contrast to make them more visible, and to add predicted rings positions as dotted lines on one half of each diffraction pattern.

3. Minor issues concerning the text

3.1. During the introduction, there is a very long sentence that is difficult to follow, starting with "However, at this ultra-small size, ...". I suggest to split it in two parts after "larger cavities than the constitutive inner ones".

3.2. Later in the introduction, the authors describe the mechanism underlying the effect of modulators in UiO-66 synthesis, and state: "modulator binds to the Zr6 nodes to produce crystals (...) with larger size due to slower nucleation". Did the authors mean "lower nucleation rate"? If so, I encourage to correct it, since a "slower nucleation" does not sound correct in this context.

3.3. The first time DMF is mentioned, the acronym is reported in the text and its meaning between brackets. In principle, it should be the opposite, and the full name should be given: "N,N-dimethylformamide (DMF)".

3.4. Some acronyms of the compounds are not appropriately introduced. For instance, in "Nitrogen porosimetry at 77K on the activated highly defective UiO-66 particles (HD-US-UiO-66) evidenced (...)" it is not directly clear that US stands for 'ultra-small'. This is not an isolated problem, so I advise the Authors to pay particular attention to where and how acronyms of the products are introduced in the manuscript.

3.5. Figure 4a is mentioned after its appearance in the text.

3.4. When commenting on the drop of performance of the products, the Authors mention a drop "to 80% of the" initial performance. Later in the text, benchmark materials are mentioned and their performance drop is expressed in terms of "activity reduction", making the comparison with the first number not direct. Ideally, both performance drops should be expressed using the same criterion, so that numbers can be compared directly.

3.6. The 'Methods' section does not seem very informative, since many more details are contained in the SI, while this section only contains a very general synthesis procedure with no characterization, etc. I do not know if this section is required and if it should be so devoid of details, but if it is, I believe it should clarify that more information can be found in the SI file.

I wish the Authors all the best with the revision of their work and I congratulate them for their remarkable results.

Kind regards

Reviewer #3 (Remarks to the Author):

In this work, the authors demonstrate that synthesis of Zr-based MOFs in water/ethanol mixtures at room temperature produces MOFs with high defect content and small crystallite sizes. Typically, this combination is hard to access using "acid modulation" which tends to produce large, defective crystals. They demonstrate the small crystallite sizes by Scherrer analysis and TEM. They demonstrate the high degree of defectiveness by TGA, elemental analysis, and surface area analysis. They demonstrate that similar conditions can be used to prepare a number of other UiO-66 derivatives, and that the defective nanocrystals have higher catalytic activities than MOFs prepared under more standard conditions. The work is carried out carefully and the authors do a great job of fully characterizing the defective materials. The paper is well-written and the results are promising for the use of Zr-MOFs in catalysis. That being said, close variations of the conditions reported here have been previously published, with an eye towards green synthesis of these materials. As a result, I do not believe the novelty of this work is sufficient to merit publication in Nature Communications and it is better suited for a more specialized journal. I have a few suggestions below that should be addressed in transferring this work to a more specialized journal.

1) The authors claim that “the surface area of 40 nm UiO-66 represents the highest value reported compared to the state-of-the-art, (1617 (\pm 5) m²/g).” This statement is not true, higher surface areas for this MOF than this have been reported: 1777 (Chem. Mater. 2016, 28, 3749–3761), 1890 (Chem. Eng. J. 2016, 289, 486–493), 1778 (Chem. Mater. 2022, 34, 3383–3394). Typically, these are obtained with TFA or benzoic acid (or benzoic acid derivatives) as modulators. The authors should amend this statement and cite these works as examples of defective MOF materials exhibiting higher surface areas. Many of these MOFs have >2 missing linkers per node (see Chem. Mater. 2022, 34, 3383–3394 Figure 6 for a meta-analysis, which is similar to Figure S11 but with many more MOF samples included), similar to the materials reported herein. Where available, the %defectiveness for these ultra-defective MOFs should be added to SI Table S2 for comparison. It would be helpful to add BET surface areas to Table S2 for comparison as well.

2) As mentioned above, various aspects of the synthetic method here have been previously reported, and in some cases, the variations on the standard method have been shown to increase the defectiveness of the formed MOFs. Researchers are certainly not good about quantifying crystallite size and defectiveness in a lot of MOF papers, but I wouldn't be surprised if some of these methods (such as room temperature synthesis in DMF, Chem. Mater. 2017, 29, 1357–1361) leads to a similar result as that reported here. The authors do a good job blending all of these approaches together to produce highly defective nanocrystals, but I do think this takes away from the novelty of their findings somewhat. In general, the authors need to make sure that this is clear in the manuscript, and the appropriate works are cited.

a) Room temperature synthesis of Zr-MOFs are known and known to increase defect content (Chem. Mater. 2017, 29, 1357–1361), presumably due to the reactions being under kinetic instead of thermodynamic control.

b) Synthesis without DMF (for example in water, to make it greener) is also well-known: see, for example, Inorg. Chem. 2015, 54, 4862–4868 (and works cited therein).

c) The synthesis of UiO-66 and derivatives from pre-formed metal oxide clusters is known (J. Am. Chem. Soc. 2023, 145, 24, 13273–13283; Inorg. Chem. 2021, 60, 14294–14301). In addition, many researchers typically pre-form these clusters under solvothermal conditions (for example, boiling ZrCl₄ and modulator together first before adding the linker). The authors should make sure to cite these works and make sure that it's clear that pre-formed clusters are known to facilitate MOF synthesis in many cases. Their use should lead to more missing linker defects than simple Zr precursors because they effectively start pre-modulated.

3) Why does lower concentration lead to more defects? (Figure 1b-c) I can understand that it has to do with the concentration of the linker in solution, but I don't see how more linker in solution leads to less growth of the crystals. Does the same effect occur in DMF? (See for example, J. Am. Chem. Soc. 2023, 145, 24, 13273–13283, Cryst. Growth Des. 2022, 22, 6379–6383).

4) The initial mention of M(IV)₆ oxoclusters is a bit vague – the authors should make it more clear that they are acetate capped clusters with similar structure as the nodes of the MOF. As mentioned above, these are known to be good precursors for MOF synthesis.

5) In the MOF synthesis procedures, the authors should make it clear at which temperature they activated the MOF under vacuum.

6) Volumes of solvents used to wash MOFs after recycling should be added to the Supporting information.

7) A more complete procedure for how the MIDA was conducted should be added to the SI.

8) The precursor Zr-oxo clusters have a PXRD pattern similar to that of the defective MOFs (Figure S25 vs Figure S1). Are the authors sure the MOFs are not contaminated by the residual clusters? Are they soluble and removed by the washing procedure?

RESPONSE TO REVIEWERS' COMMENTS

Reviewer #1 (Remarks to the Author):

Dai, Tissot, Parac-Vogt, Serre and co-workers report the methodology for the synthesis of Zr and Hf containing MOF nanocrystals with a high percentage of defects. The work was mainly performed on the UiO-66 Zr(IV) system (and MOF-801) where the generation of defects have previously been well characterised with increasing concentrations of modulators (monocarboxylic acids) used. The generation of MOF nanocrystals with a high percentage of defects is of importance for catalysis where higher reactivity can be achieved. The research is highly topical, relevant to numerous applications and an important contribution to the MOF literature. The work will be of interest to the broad readership of Nature Communications. I thoroughly enjoyed reading the manuscript - all experiments were performed to a high level and conclusions made from experimental work are sound. The mechanism of MOF nanocrystal formation and the role of the solvent were well considered.

Response: We thank very much the referee for this highly positive comment and for his/her time in reviewing our work.

I only had a minor comment regarding the yields obtained from the synthesis of the MOF nanocrystals - could these please be included in the experimental section? Does the yield change the size of particle obtained or with the substituent on the terephthalate ligand used for the synthesis of UiO-66? The synthesis of the MOF nanocrystals are performed at a small scale (75 mg of the starting Zr cluster) - is it possible to scale up the synthesis?

Response: Thanks for mentioning this very important point. The product yields of 4-6 nm UiO-66-X are not mentioned in the paper because it is really difficult to give an accurate value. As is shown in Figure 4a (main text) and Figure R1, the colloidal stability of highly defective UiO-66-X in synthetic solution is extremely good, which prevents us from effectively recovering all the nanoparticles from the solution. The approach proposed in our paper was based on slow rotary evaporation at room temperature and a subsequent high-speed centrifuge at 14,500 rpm. The yield, based on Zr, was often around 50-60%. Indeed, we have also tried to use a higher-speed centrifuge (up to 18,000 rpm), and a higher yield was clearly obtained (ca. 75%). However, we observed a considerable amount of MOF powder after fully evaporating the supernatant recovered after centrifugation, which evidences that none of these techniques are able to fully recover the nanoparticles from the solution. The change of organic linker from terephthalate to other substituents only makes it harder as the resulting MOFs are often more hydrophilic and thus more soluble than pristine UiO-66.

In another experiment, we have fully evaporated the synthetic solution and measured the PXRD pattern of the collected product. It is pretty clear there is no significant diffraction peak of terephthalic acid (Figure R2), which means the vast majority of linker is coordinated to the metal ions, which is in agreement with an almost quantitative reaction.

Thus, we have added a new statement in the SI to highlight this point.

'Note: the yield of HD-US-UiO-66-X synthesis is highly dependent on the isolation process, particularly of the centrifugation conditions, as it is extremely difficult to fully recover these colloidal stable MOF nanoparticles from the solution. With the above-described process, the yield is around 50-60%. However, this can be improved to 75% by using an ultrahigh-speed centrifuge (18,000 rpm). Depending

on the instrument accessibility, we strongly suggest using centrifuge speed as high as possible to obtain the highest product yield.'

Figure R1. Hydrodynamic size of HD-US-Uio-66 colloids ($T = 25\text{ }^{\circ}\text{C}$) determined by in situ time-dependent DLS (time resolution $t_R = 120\text{ s}$)

Figure R2. The PXRD patterns ($\lambda_{Cu} = 1.5418\text{ \AA}$) of the synthesized HD-US-Uio-66 recovered after complete evaporation of the solvent compared with the one of terephthalic acid.

In addition, we believe that our synthesis, including the preparation of Zr_6 oxoclusters, is potentially scalable as it is performed using milder conditions than reflux synthesis, that is often considered as a scalable method.

Reviewer #2 (Remarks to the Author):

Dear Authors and Editor

This manuscript reports a novel synthetic approach for the production of Zr- or Hf-based UiO-66 nanoparticles. The method proposed by the Authors allows for a useful control over the size of the produced particles down to a remarkably small length of less than 10 nanometers, which is undoubtedly of great interest for MOF applications traditionally hampered by mass transport limitations (e.g. heterogeneous catalysis). The study also shows some synthetic influence over the type and amount of defects, although their characterization—notoriously challenging—lacks the level of details and accuracy for substantiating some of the Authors' claims. Last but not least, the proposed synthetic strategy is more sustainable and environmentally friendly than most MOF syntheses, which still largely rely on the use of N,N-dimethylformamide and lengthy solvothermal conditions. After describing a reasonably thorough characterization of the products and providing some useful—although mostly speculative—considerations on their formation, the Authors conclude by testing the versatility of their method on differently functionalized forms of UiO-66, and by

assessing the catalytic performance of some of them compared to forms of the same materials having different crystallinity.

Overall, I consider this work of high interest for the community and I am positive its quality and novelty qualify it for publication in Nature Communications. However, I believe some parts of this study lack clarity, information, or contain possibly incorrect statements. Therefore, a revision is essential to make it suitable for acceptance and publication. My concerns and suggestions are reported as follows.

Response: We thank the referee for his/her positive remarks as well as for his constructive comments that allowed us to improve the quality of our manuscript.

1. Scientific concerns

1.1. At the beginning of the "Results" section, the authors classify the ultra-small and highly defective UiO-66 particles as "highly crystalline". This is quite confusing, since—in simple terms—crystallinity is the degree of translational order in a material (direct space definition) or the 'sharpness' of peaks in their diffraction pattern (reciprocal space definition; <https://dictionary.iucr.org/Crystal>). The confusing aspect is that this entire study focuses on achieving the lowest possible crystallinity (smallest crystal size, highest amount of defects). The tendency of referring to these products as 'highly crystalline' is, in my view, part of a general phenomenon leading reticular chemists to claim their compounds are 'highly crystalline' as if this was a must-have virtue for a material to be useful. Needless to say that this is not based on reality, and the properties of highly defective and ultra-small UiO-66 particles prove further that low crystallinity is sometimes extremely interesting. It is, as a matter of fact, what the Authors pursue in their work. I would therefore encourage them to revise their manuscript by re-thinking the best phrasing of all sentences where the concept of 'crystallinity' or 'crystalline' is used, since this term is so often abused and misused in the MOF&COF community.

Response: We agree with the referee that it can be contradictive to use 'highly crystalline' to describe the materials that does not diffract much. However, HRTEM and 77K N₂ adsorption data both evidenced that the particles present a high degree of translational order in the direct space, so we believe they can still be considered as crystalline according to the definition given by the referee. In order to avoid any misleading, we have replaced 'highly crystalline' by 'high quality' in the manuscript.

1.2. Still in the very first line of the "Results" section, the authors invoke the concept of "maximal defect content". Is it certain that the maximal content of defects is achieved, and particles with more defects are theoretically impossible to form? If so, the authors should demonstrate this point since, in my view, it is a rather strong and possibly inaccurate claim.

Response: The referee is right. We can't demonstrate here if the defect content has reached the maximum level as this will need perhaps theoretical calculation, etc. We have corrected our description to 'To prepare high quality ultra-small nanoparticles of UiO-66(Zr) with high defect content, we first considered carefully the main relevant state-of-the-art strategies.'

1.3. When the Authors describe that the particle size value of 44 nm calculated by the Scherrer equation is very close to that observed by TEM, they conclude that the particles are well-crystalline. I do not see the connection here. The most obvious conclusion that one could draw is that, if we consider the Scherrer equation method reliable, the sample observed by TEM has been fairly representative of the bulk. The fact that the Scherrer equation produces results matching the observation by TEM is not a sign that the particles are well-crystalline, especially since the Scherrer equation itself shows that, as desired, the crystallinity is relatively low: a few tens of unit cells down to less than ten.

Response: Thanks for pointing out this description. We agree with the referee that the results from Scherrer equation and TEM observation are not directly related to the crystal quality. However, the matching of Scherrer equation and TEM indicates that, in general, the nanoparticles observed are built with a single crystal domain and not from aggregation of domains of smaller size, which is linked to the quality of the sample. We have modified the sentence accordingly in order to be more accurate: 'This

value is close to the particle size (44 nm) calculated from Scherrer equation, indicating that the particles are mainly single crystal domains.'

1.4. The Authors mention that the "absence of uncoordinated carboxylic acid residual groups in the washed materials" is "in agreement with the presence of linker defects". I cannot see the link between the two aspects. UiO-66 particles that are free of uncoordinated carboxylic groups can still be fully-connected structures without any defect. I believe that either the authors created an incorrect correlation between IR-based evidence and presence of linker defects, or I did not correctly understand their statement. In the latter case, I kindly ask the authors to make this sentence clearer.

Response: The referee is correct. UiO-66 that is free of uncoordinated carboxylic acid groups can be intact UiO-66. However, in this study, we have observed BET surface areas that are larger than the theoretical ones, which is likely due to the presence of defects that could be missing linkers or missing oxoclusters. In this context, we applied FTIR, in addition to PXRD and HRTEM, to discriminate between both hypotheses, as missing oxoclusters should lead to uncoordinated carboxylic acid groups in the structure while missing linkers should not. We have modified the text to clarify this point: 'demonstrated the absence of uncoordinated carboxylic acid residual groups in the washed materials, in agreement with the presence of linker defects rather than missing oxoclusters.'

1.5. When discussing the TEM imaging results of the smallest particles, the Authors correlate being able to resolve all clusters in the images to having missing linker defects. I don't see how this link is sensible. From the available TEM images there is no direct information on the presence or absence of linkers and it is therefore not possible to make any conclusion concerning linker defects. Moreover, especially with respect to these small particles, deviations in cluster-to-linker ratios are not necessarily indicative of missing linkers in the framework internal structure. Indeed, if we consider a 2x2x2 unit cells UiO-66 particle and hypothesize that modulators acted as capping agents to obtain a fully cluster-terminated particle, its composition would be 63 clusters and 240 linkers when all linkers are present. This would mean a linker-Zr₆ ratio of ca. 3.809 even without defects within the structure. Since the ratio that leads the Authors to claim that linkers are missing is actually higher (ca. 3.9), it could mean that these hypothetical 2x2x2 particles, although having no defects, have an excess of linker—which can only be attached to their outside. Of course, these considerations become less relevant for larger particles, where the evidence for missing linker defects is stronger since the surface termination plays a weaker role in defining the linker-cluster ratio. Nevertheless, I encourage the Authors to invest some efforts in commenting these aspects, thereby adding some healthy caution that I am sure would be highly appreciated by most readers.

Response: Thanks for pointing out this important point. We agree with the fact that it is not possible to observe missing linker defects according to our HRTEM image. However, the individual Zr₆ oxoclusters are electron-rich enough to be clearly observed in the HRTEM images. The results shown in Figure 2.i and f are indicative of a lack of Zr₆ oxocluster defects, which in turn implies the missing linker defects are more plausible in our case (in view of the other characterization techniques evidencing the presence of defects in the particles). This point has been clarified now in the manuscript as follows: 'From the profile analysis on HRTEM images along (220) and (011) directions (Figure 2f), the distances between two adjacent Zr₆ oxoclusters are highly homogeneous with an average value of 1.1 nm that is close to the theoretical one (1.2 nm), suggesting the absence of oxocluster defects in our ultrasmall UiO-66 (HD-US-UiO-66).'

2. Missing information

2.1. In the introduction, the Authors discuss the advantages of downsizing MOFs, providing useful examples of "enhanced properties (i.e., catalysis, sensing, biomedicine,...)". Right afterwards, the authors also mention "novel features", although it is not clear what these features are, and no reference is provided. It would be useful to mention these novel features between brackets and provide appropriate references to make this addition meaningful.

Response: Concerning the 'novel features' many things can be envisioned and we have cited several relevant references following this statement 'novel feature' in the original manuscript. The references 8, 12, and 13 are relevant examples of the new features researchers have discovered using nano-MOFs,

including completely different flexible behaviour and/or different optical properties. This is now explicitly mentioned in the introduction *'The reduction of the MOFs size to the nanoscale has imparted nanoMOFs with various enhanced properties (i.e., catalysis, sensing, biomedicine...)^{8, 9, 10, 11} and novel features (flexibility, optical properties)^{8, 12, 13} but despite advances, the design of ultra-small MOF nanoparticles still faces severe difficulties.^{14'}*

2.2. Later in the introduction, the Authors report how the MIDA approach has been mainly used on "UiO-66 with missing linker defect content of ca. 10-20%." This statement requires at least one reference.

Response: Thanks to the referee for this reminder. We have added a few references following to this statement. *'The MIDA approach, usually carried out in toxic N,N-dimethylformamide (DMF) at high temperature and pressure, has mainly been explored with UiO-66 with missing linker defect content of ca. 10–20%.^{18, 34, 35'}*

2.3. Missing clusters defects were deemed absent based on XRD profiles, as "no symmetry-forbidden peak at low-angle" were observed. The authors mention that these peaks should be visible below $2\theta = 7^\circ$, but it would be most informative to mention at which 2θ values these extra peaks are expected, since this is a piece of information that can be easily provided.

Response: We have modified our statement in the main text as following. *'No symmetry-forbidden peaks at low angle (2θ at ca. 4 and 6 $^\circ$) were observed in the PXRD pattern of the sample, suggesting the absence of missing cluster defects that would lead to an ordered structure with reo topology (Figure S2).*

2.4. In Figure 2, the scale bar of the diffraction pattern is reported in pixel instead of meaningful units such as $1/\text{nm}$ or $1/\text{Å}$. Please insert a correct scale bar for this picture.

Response: Thanks for reminding us this important point. We have corrected the scale bar in the main text, as shown here in Figure R3.

Figure R3. SAED pattern of the 5 nm UiO-66.

2.5. Still in Figure 2, in the stacked diffraction pattern the label of the last reflection appears to be wrong. Not only the (0 2 2) reflection is already reported to its left, but UiO-66 has two different reflections in this region: (1 1 3) and (2 2 2). Which one did the Authors intend to highlight?

Response: We thank the reviewer for pointing out this mistake. We have corrected accordingly the reflection label in the new figure 2b (and shown in Figure R4).

Figure R4. powder X-ray diffraction (PXRD) ($\lambda_{\text{Cu}} = 1.5406\text{\AA}$) patterns of UiO-66 synthesized with different volumes of EtOH.

2.6. I understood that the linker, once added in the syntheses of the nanoparticles, is not fully dissolved but instead forms a suspension. I think this is very important to state more explicitly, since this would mean that the growth of UiO-66 nanoparticles might proceed via heterogeneous nucleation on the undissolved linker particulate, and also that linker solubility strongly influences the kinetics of the particles' growth.

Response: We are not convinced that the UiO-66 nanoparticles grow through heterogeneous nucleation as the linkers shall be dissolved in order to coordinate with the metal cations. However, it is true that terephthalic acid is not fully dissolved in EtOH at the beginning at RT when the solvent volume is low. We believe that as the MOF nucleation-growth proceeds, the linker is getting progressively solubilised as the equilibrium is displaced. This aligns with our observations where the introduction of more EtOH or of DMF in the solution leads to faster MOF nucleation-growth (thus smaller size) because of the higher linker solubility.

2.7. The Authors mention something interesting: "although Metal–Organic Cages/Polyhedra (MOCs/MOPs) have intrinsically discrete supramolecular architectures, they often suffer strongly from poor chemical/hydrolytic stabilities and structural collapse upon activation". Once having read this, I immediately wondered how this transfers to such small $2 \times 2 \times 2$ unit cell particles or analogous size. This appears to be implied by the final sentence, but the point is not clarified further, leaving the readers wondering: is this true also for these ultra small particles when their size is extremely small? To which extent? Does the fact that this is not strictly a cage, but a slightly larger coordination assembly, change these considerations? And does it change them only quantitatively or also qualitatively? I believe that, for the sake of being informative and transparent, the Authors should briefly address these aspects when concluding this paragraph.

Response: The poor chemical/hydrolytic stabilities of many MOCs/MOPs originate from their weak assembly and coordination bonds. In our case, the coordination bond between Zr and carboxylate is stronger than in most of the reported MOPs and MOCs. In addition, the face-centered unit of UiO-type MOFs offers them extraordinary thermal/chemical stability even with the presence of defects compared with so many Zr-based MOFs. In this study, we have performed in situ variable temperature PXRD and catalytic reaction (under acidic condition, *ca.* pH=4). These results have demonstrated the good thermal/chemical stabilities of our HD-US-UiO-66, even though this compound is slightly less stable than larger particles. For example, a decrease in thermal stability was observed (up to 300 °C) compared to the value of UiO-66 in a few hundred nanometers (up to 500 °C, DOI: 10.1021/ja8057953). However, we believe the stability shown here is sufficient for many low temperature catalytic/optical applications, including the ones demonstrated in our paper. The manuscript has been modified to clarify this statement: 'However, in contrast with our ultrasmall particles (see after for the description of the thermal and chemical stability), Metal–Organic Cages/Polyhedra (MOCs/MOPs) often suffer strongly from poor chemical/hydrolytic stabilities and structural collapse upon activation, preventing their applications.'

2.8. When rationalizing the formation of HD-US-UiO-66, the Authors mention a pH condition of ca. 4. It is not clear where this value originates from, especially since it is not mentioned in the synthetic procedures (SI). Probably, this value refers to a previous study, in which case the Authors are kindly advised to add a dedicated reference.

Response: We have measured the pH value ourselves. The relevant information is now added in the SI. 'The pH value of Zr₆ acetate oxoclusters solutions were measured in 5 mL, 20 mL, and 80 mL of water with 0.3 g Zr₆ acetate oxoclusters, giving values between 3.6 and 4.4.'

2.9. The Authors mention an "excellent colloidal stability" assessed by in situ TD-DLS. Is there a most-likely origin of this stability? If a sound hypothesis could be formulated, it would be interesting if the Authors could briefly introduce it after the stability assessment.

Response: The excellent colloidal stability shown by the in-situ TD-DLS can be explained by the highly charged nature of the synthesized MOF nanoparticles. The Zeta potential of the HD-US-UiO-66 (Figure S21) in water was ca. +40 mV, which leads to strong electrostatic repulsion between the nanoparticles in water. This point has been clarified in the manuscript: 'This stability, correlated to the highly positive charge evaluated by Zeta potential analysis, was observed whatever the nanocrystal size'.

2.10. There is some useful information missing in the experimental procedure reported in the SI, as follows.

- In the preparation of the Zr₆ oxoclusters it is not clear how the chemical composition of the clusters has been determined. If this refers to a past study, it should be stated explicitly.

Response: Yes, the chemical composition of Zr₆ oxoclusters has been previously determined in other paper. In the SI, this has been mentioned and cited. Please see the synthesis section in the page 2 of SI.

- Essentially every synthesis procedure mentions a "completely colorless" reaction mixture. Since I assume the reactants are colorless to begin with, I guess the Authors meant "clear" and not "colorless".

Response: We appreciate the referee for reminding us this. We have corrected the all the wrong description in the paper and highlighted them.

- The appearance of the solution after linker addition is not followed by whether this resulted in a clear solution or in a suspension. From what I understood, poor linker solubility often resulted in suspensions being formed. This should be mentioned explicitly since it is a critical piece of information to better understand the conditions under which the products are formed.

Response: The referee is right. We have modified our statement in the paper. '*Ethanol was introduced into the solution followed by the immediate addition of 1.2 mmol BDC-X, and the suspension was stirred for 2h at room temperature. Note, with the reaction proceeds, the suspended linkers were fully dissolved after 2h synthesis.*'

- Several synthesis procedures report a volume of acetic acid without clarifying its concentration. Please add this piece of information.

Response: We have added this information now in the SI. Please see here an example: '*Zr₆ oxoclusters (0.06 mmol, 75 mg) were dispersed in acetic acid (0.5 mL, 8.75 mmol) under stirring at 600 rpm. H₂O (1.25 mL) was subsequently added, and the reaction mixture was stirred until it became completely clear. 10 mL of ethanol was introduced into the solution followed by the immediate addition of 1,4-benzenedicarboxylic acid (BDC, 50 mg, 0.3 mmol), and the reaction was stirred for 3h at room temperature. The resulting solution was centrifuged at 14,500 rpm for 45 min and then washed twice with ethanol (14,500 rpm, 45 min). The collected solid was dried under vacuum for 3 h for characterizations and applications.*'

2.11. In the SI Figure S26, the images show some ring patterns that are extremely difficult to distinguish from the background. I advise the Authors to change intensity scale and contrast to make them more visible, and to add predicted rings positions as dotted lines on one half of each diffraction pattern.

Response: We thank the referee for this point. Due to the fact that nanoparticles concentration used for TEM imaging was quite low, we were unable to get better quality SAED. However, the electron diffraction rings in these patterns agree well with the Figure 2e in the main text, indicating the same synthesized structures. The diffraction rings have now been added in the Figure S26.

3. Minor issues concerning the text

3.1. During the introduction, there is a very long sentence that is difficult to follow, starting with "However, at this ultra-small size, ...". I suggest to split it in two parts after "larger cavities than the constitutive inner ones".

Response: We have split this long sentence to two sentences in the main text. 'However, at this ultra-small size, the majority of the atoms of MOFs lie close to the external surface, decorated with larger cavities than the constitutive inner ones. This maximizes the interface for substrate interaction alongside largely decreased diffusion/desorption path length,⁴ naturally resulting in enhanced catalytic properties.'

3.2. Later in the introduction, the authors describe the mechanism underlying the effect of modulators in UiO-66 synthesis, and state: "modulator binds to the Zr₆ nodes to produce crystals (...) with larger size due to slower nucleation". Did the authors mean "lower nucleation rate"? If so, I encourage to correct it, since a "slower nucleation" does not sound correct in this context.

Response: The referee is right. We have corrected our statement in the main text. '*During the MOF synthesis, the modulator binds to the Zr₆ nodes to produce crystals with lower connectivity, and consequently with larger size due to lower nucleation rate and crystal growth kinetics.*'

3.3. The first time DMF is mentioned, the acronym is reported in the text and its meaning between brackets. In principle, it should be the opposite, and the full name should be given: "N,N-dimethylformamide (DMF)".

Response: Thanks for this careful check. We have changed it in the main text as following. 'The MIDA approach, usually carried out in toxic N,N-dimethylformamide (DMF) at high temperature and pressure.'

3.4. Some acronyms of the compounds are not appropriately introduced. For instance, in "Nitrogen porosimetry at 77K on the activated highly defective UiO-66 particles (HD-US-UiO-66) evidenced (...)" it is not directly clear that US stands for 'ultra-small'. This is not an isolated problem, so I advise the Authors to pay particular attention to where and how acronyms of the products are introduced in the manuscript.

Response: Thanks for reminding us this point. We have paid attention to the acronyms in the paper and made some corrections. Here is one example: '*From the profile analysis on HRTEM images along (220) and (011) directions (Figure 2f), the distances between two adjacent Zr₆ oxoclusters are highly homogeneous with an average value of 1.1 nm that is close to the theoretical one (1.2 nm), suggesting that the defects are missing linkers in our highly defective ultrasmall UiO-66 (HD-US-UiO-66).*'

3.5. Figure 4a is mentioned after its appearance in the text.

Response: We have moved the position of Figure 4 in the manuscript.

3.4. When commenting on the drop of performance of the products, the Authors mention a drop "to 80% of the" initial performance. Later in the text, benchmark materials are mentioned and their performance drop is expressed in terms of "activity reduction", making the comparison with the first number not direct. Ideally, both performance drops should be expressed using the same criterion, so that numbers can be compared directly.

Response: We have modified the sentence and made the comparison more direct. Please see below the new sentence. '*This is however still an excellent recyclability, especially when compared to benchmark materials, such as MIP-201 and MOF-808,^{48, 49} which suffered from lower recyclability (only 45% and 20% activity after 5 cycles, respectively, Figure 5d),*'

3.6. The 'Methods' section does not seem very informative, since many more details are contained

in the SI, while this section only contains a very general synthesis procedure with no characterization, etc. I do not know if this section is required and if it should be so devoid of details, but if it is, I believe it should clarify that more information can be found in the SI file.

Response: We agree with the referee that the SI is more informative than the 'Method' section in the main text. After checking the Nature Communications formatting instructions, we have deleted this section in the main text and moved it to the SI.

I wish the Authors all the best with the revision of their work and I congratulate them for their remarkable results.

Response: We are very grateful for all these highly constructive/useful comments the referee made.

Reviewer #3 (Remarks to the Author):

In this work, the authors demonstrate that synthesis of Zr-based MOFs in water/ethanol mixtures at room temperature produces MOFs with high defect content and small crystallite sizes. Typically, this combination is hard to access using "acid modulation" which tends to produce large, defective crystals. They demonstrate the small crystallite sizes by Scherrer analysis and TEM. They demonstrate the high degree of defectiveness by TGA, elemental analysis, and surface area analysis. They demonstrate that similar conditions can be used to prepare a number of other UiO-66 derivatives, and that the defective nanocrystals have higher catalytic activities than MOFs prepared under more standard conditions. The work is carried out carefully and the authors do a great job of fully characterizing the defective materials. The paper is well-written and the results are promising for the use of Zr-MOFs in catalysis. That being said, close variations of the conditions reported here have been previously published, with an eye towards green synthesis of these materials. As a result, I do not believe the novelty of this work is sufficient to merit publication in Nature Communications and it is better suited for a more specialized journal. I have a few suggestions below that should be addressed in transferring this work to a more specialized journal.

1) The authors claim that "the surface area of 40 nm UiO-66 represents the highest value reported compared to the state-of-the-art, (1617 (\pm 5) m²/g)." This statement is not true, higher surface areas for this MOF than this have been reported: 1777 (Chem. Mater. 2016, 28, 3749–3761), 1890 (Chem. Eng. J. 2016, 289, 486–493), 1778 (Chem. Mater. 2022, 34, 3383–3394). Typically, these are obtained with TFA or benzoic acid (or benzoic acid derivatives) as modulators. The authors should amend this statement and cite these works as examples of defective MOF materials exhibiting higher surface areas. Many of these MOFs have >2 missing linkers per node (see Chem. Mater. 2022, 34, 3383–3394 Figure 6 for a meta-analysis, which is similar to Figure S11 but with many more MOF samples included), similar to the materials reported herein. Where available, the %defectiveness for these ultra-defective MOFs should be added to SI Table S2 for comparison. It would be helpful to add BET surface areas to Table S2 for comparison as well.

Response: We are grateful for reminding us these important points. The surface area reported in these articles (ca. 1800 m²/g) is indeed higher than the value in our paper.

1. We have thus amended the statement in the main text. We used the high surface area to indicate the high quality, as well as high defectiveness of our synthesized 40 nm UiO-66. The references mentioned by the referee have been cited: *'the surface area of 40 nm UiO-66 represents a comparably high value compared to the state-of-the-art.'*
2. The referee is right. Typically, the use of either very strong modulator (TFA) or larger-sized aromatic acid (Benzoic acid and derivatives) could lead to materials with a large number of defects. In these references (for example, Chem. Mater. 2022, 34, 3383–3394), it has been clearly pointed out in the main text (quoted below) that although a strongly acidic combined to a larger-sized modulator (thiophenecarboxylic acid, 3-TP) could be very effective in producing larger crystals (up to 1 μ M) with highest surface area, the defect content is even higher than the smaller crystalline domains (< 200nm). This observation agrees well with our research motivation, where we are aiming to maintain the high defectiveness of MOF when downsizing.

‘Critically, we did not observe any samples that possess large crystalline domains and low amounts of modulator incorporated as defects (upper right section of Figure 6). In samples with smaller crystalline domains (empirically determined as LVol-IB <200 nm), the extent of linker incorporation generally skews toward a higher number of linkers incorporated compared to modulators (Figure 6 and Figures S4 and S5), possibly due to the weaker ability of their associated modulators to compete with terephthalate. Overall, these findings indicate that a high degree of defectiveness does not imply high crystallinity, but samples with large crystalline domain sizes do tend to possess a large degree of linker substitution defects due to coordinative displacement of the linker by the modulator.’

3. We don't think the figure 6 in Chem. Mater. 2022, 34, 3383–3394 is a meta-analysis of the amount of defects observed in different articles. It only shows the distribution of defect content/size from different synthesis batches prepared for this particular publication
4. We agree with the referee it is better to add the % of defectiveness of our materials and the relevant surface area in the table S2 for comparison. Please check the revised table below.

Table S2. Summary of reported works that dealt with defect-engineering on UiO-66.

MOF	Missing linker/ %	Size	Surface area (m ² /g)	Reference
UiO-66	27%	7 μm	NA	3
UiO-66	16%	300-600 nm	NA	4
UiO-66	29%	300 nm	1520	5
UiO-66	5%	10 nm	1130/1250	5
UiO-66	10%	120 nm	1343	6
UiO-66	17%	250 nm	1391	6
UiO-66	29%	1000 nm	1479	6
UiO-66	16%	200 nm	1546	7
UiO-66-NDC	0.3%	100 nm	1272	8
UiO-66-NDC	12.6%	740 nm	1319	8
UiO-66	5%	20 nm	1376	9
UiO-66	4%	20nm	700	10
UiO-66 (HD-US-UiO-66-X)	35%	4-6 nm	428-1040	This work

2) As mentioned above, various aspects of the synthetic method here have been previously reported, and in some cases, the variations on the standard method have been shown to increase the defectiveness of the formed MOFs. Researchers are certainly not good about quantifying crystallite

size and defectiveness in a lot of MOF papers, but I wouldn't be surprised if some of these methods (such as room temperature synthesis in DMF, Chem. Mater. 2017, 29, 1357–1361) leads to a similar result as that reported here. The authors do a good job blending all of these approaches together to produce highly defective nanocrystals, but I do think this takes away from the novelty of their findings somewhat. In general, the authors need to make sure that this is clear in the manuscript, and the appropriate works are cited.

Response: We believe our study represents an original approach compared to the literature for several reasons:

In terms of synthesis method, we have proposed an entirely green ambient pressure approach to synthesize benchmark Zr-MOFs. Consequently, the controllable synthesis allows us to prepare nano-MOFs with only a few unit cells, which is certainly close to the limitation of such 'crystalline' materials. In addition, in contrast to the previous studies, where the defect content is strongly dependent on the modulator quantity, we found that the defect number can be kept constant by simply controlling the kinetics of crystallization. In addition to the production of many UiO-type MOFs with similar size and defect, this study provides a new understanding of the kinetics control at room temperature.

In terms of applications, this paper demonstrates the benefits of these ultrasmall and highly defective Zr-MOFs toward a much better peptide hydrolysis/condensation. We have found that, by significantly reducing the size of porous catalysts and increasing the defect number, the catalytic efficiency, mass transfer efficiency, and reaction flexibility are all improved to a top-performing level without sacrificing the selectivity and stability. All these discoveries have set up a new benchmark for such catalytic reaction and will encourage more studies in the future.

In terms of the potential properties of the ultrasmall highly defective Zr-MOFs, we have shown exceptional optical properties of our materials as evidenced by in situ TD-DLS (Figure 4a, Figure S20) and luminescence measurements (Figure S24). The colloidal stability and stable optical properties are prerequisites for many sensing, MOF-based MMMs, and biological applications.

In short, this paper shall be of interest for researchers working on nano/defective-MOF synthesis, heterogeneous catalysis, and many other relevant applications and we therefore believe that it deserves publication in Nature Communications.

a) Room temperature synthesis of Zr-MOFs are known and known to increase defect content (Chem. Mater. 2017, 29, 1357–1361), presumably due to the reactions being under kinetic instead of thermodynamic control.

Response: The paper mentioned by the referee (Chem. Mater. 2017, 29, 1357–1361) clearly reported that defect content increases with the reduction of temperature (down to room temperature, Figure 6_{ref}. Maximum defect, 20%). Moreover, their room temperature UiO-66 showed an average size at 250 nm (SEM, Figure S1_{ref}), which means our synthesis method reported here has led to significantly different defect content and nanoparticle size (35-40%, 4-6 nm).

b) Synthesis without DMF (for example in water, to make it greener) is also well-known: see, for example, Inorg. Chem. 2015, 54, 4862–4868 (and works cited therein).

Response: This work mentioned by the referee reported the preparation of UiO-66-X bearing a large particle size and unknown defect content, in water at 100 °C with scalability which, we believe, is quite different from our story for the reasons explained before.

c) The synthesis of UiO-66 and derivatives from pre-formed metal oxide clusters is known (J. Am. Chem. Soc. 2023, 145, 24, 13273–13283; Inorg. Chem. 2021, 60, 14294–14301). In addition, many researchers typically pre-form these clusters under solvothermal conditions (for example, boiling ZrCl₄ and modulator together first before adding the linker). The authors should make sure to cite these works and make sure that it's clear that pre-formed clusters are known to facilitate MOF synthesis in many cases. Their use should lead to more missing linker defects than simple Zr precursors because they effectively start pre-modulated.

Response: It has been demonstrated in the previous works, including the ones the referee exemplified in the comments, that the use of other inorganic Zr salts (ZrCl₄) can also lead to highly defective Zr-

MOFs. However, as we have pointed out in the manuscript, the use of highly acidic $ZrCl_4$ or $ZrOCl_2$ can modulate the formation of corresponding MOFs due to the inhibited coordination between carboxylic acid and Zr. In this case, we can expect to achieve Zr-MOFs with higher defect content while, often, larger size. We agree with the fact that preformed oxoclusters are making the MOF synthesis easier but we are not sure if the preformed Zr oxoclusters could facilitate the formation of missing linker.

3) Why does lower concentration lead to more defects? (Figure 1b-c) I can understand that it has to do with the concentration of the linker in solution, but I don't see how more linker in solution leads to less growth of the crystals. Does the same effect occur in DMF? (See for example, J. Am. Chem. Soc. 2023, 145, 24, 13273–13283, Cryst. Growth Des. 2022, 22, 6379–6383).

Response: We have clearly shown in the manuscript that lowering the reactant concentration leads to a decrease of the MOF particle size while maintaining constant the defect content. Moreover, we have evidenced that increasing the linker solubilization does accelerate the MOF growth and consequently lead to smaller nanoparticles. We believe the two references are not so relevant to our paper as they only discussed the Zr/Cu-MOF synthesis dealt with high concentration method and seeding method.

4) The initial mention of $M(IV)_6$ oxoclusters is a bit vague – the authors should make it more clear that they are acetate capped clusters with similar structure as the nodes of the MOF. As mentioned above, these are known to be good precursors for MOF synthesis.

Response: We apologize for the vague description of Zr_6 oxoclusters. Actually, we have cited the corresponding original paper for the synthesis and characterizations of Zr_6 acetate clusters in the SI. To make it clearer, we have now mentioned this point in the main text: *'The initial synthesis of UiO-66(Zr) was performed by first mixing acetate capped Zr_6 oxoclusters with acetic acid. Water, ethanol, and benzene-1,4-dicarboxylic acid (BDC) were subsequently introduced in the oxocluster solution (see detail in SI).'*

5) In the MOF synthesis procedures, the authors should make it clear at which temperature they activated the MOF under vacuum.

Response: We thank the referee for reminding us this important point. We have actually mentioned the activation temperature in the instruments section in SI. *'Nitrogen porosimetry data were collected on a Micromeritics Tristar/ Triflex instrument at 77 K (pre-activating samples at 70 °C under vacuum, 10 hours).'* To make it clearer, we have added a new section in the SI to remind readers this parameter: *'Each 77K N2 isotherm was using around 50-100 mg of MOF powder. The MOF powder was first loaded in a glass tube and thermally activated at 70 °C under dynamic vacuum for 10 hours before the measurement.'*

6) Volumes of solvents used to wash MOFs after recycling should be added to the Supporting information.

Response: The volumes of solvents used to wash MOFs during each cycling experiments are now given in the SI. Please see the updated statement: *'For recycling experiments, after reaction, the MOFs were washed in D2O overnight to remove adsorbed substrate and product, followed by washing in 10 mL of acetone and drying in the oven for 8 h at 110 °C. After which the reaction was repeated.'*

7) A more complete procedure for how the MIDA was conducted should be added to the SI.

Response: Thanks for reminding us this point. Actually, the synthesis of modulator-induced-defect approach (MIDA) has been described in the SI.

'Synthesis of MI-US-UiO-66 (5 nm)

Zr_6 oxoclusters (0.06 mmol, 75 mg) were dispersed in acetic acid (0.025 mL) under stirring at 600 rpm. H_2O (1.25 mL) was subsequently added, and the reaction mixture was stirred until it became completely clear. 10 mL of ethanol was introduced into the solution followed by the immediate addition of 1,4-benzenedicarboxylic acid (BDC, 50 mg, 0.3 mmol), and the reaction was stirred for 1h at room temperature. The resulting solution was centrifuged at 14,500 rpm for 45 min and then washed twice with ethanol (14,500 rpm, 45 min). The collected solid was dried under vacuum for 3 h for

characterizations and applications. Modulator induced size control can be applied with different amount of acetic acid used in the synthesis batch.'

8) The precursor Zr-oxo clusters have a PXRD pattern similar to that of the defective MOFs (Figure S25 vs Figure S1). Are the authors sure the MOFs are not contaminated by the residual clusters? Are they soluble and removed by the washing procedure?

Response: We thank the referee for this comment. The relevant PXRD patterns have been plotted together and shown below in Figure R5. We didn't see any diffraction peaks on our synthesized MOFs that can be attributed to the Zr₆ oxoclusters. In addition, as mentioned in the synthesis description, the Zr₆ oxoclusters are very soluble in the presence of H₂O, indicating they can easily be removed by simple washing.

'Zr₆ oxoclusters (0.06 mmol, 75 mg) were dispersed in acetic acid (0.5 mL, 8.75 mmol) under stirring at 600 rpm. H₂O (1.25 mL) was subsequently added, and the reaction mixture was stirred until it became completely clear. 80 mL of ethanol was introduced into the solution followed by the immediate addition of 1,4-benzenedicarboxylic acid (BDC, 50 mg, 0.3 mmol), and the reaction was stirred for 2h at room temperature.'

Figure R5. The PXRD patterns of HD-US-UiO-66-X (X=NH₂, NO₂, Br, (OH)₂) in comparison to simulated UiO-66 and Zr₆ oxoclusters.

REVIEWERS' COMMENTS

Reviewer #1 (Remarks to the Author):

The authors have addressed my concerns in a thoughtful and thorough manner. I recommend acceptance of the manuscript.

Reviewer #2 (Remarks to the Author):

Dear Authors

I am satisfied with the thorough revision that has been made to address my points, as well as with the requested clarifications.

I am glad to support the publication of this study.

Kind regards

Reviewer #3 (Remarks to the Author):

The authors have done an excellent job of addressing all of the reviewer concerns. I believe this work is now suitable for publication in Nature Communications.

RESPONSE TO REVIEWERS' COMMENTS

Reviewer #1 (Remarks to the Author):

The authors have addressed my concerns in a thoughtful and thorough manner. I recommend acceptance of the manuscript.

Response: We thank the referee for this very positive comment and for his/her time in reviewing our work.

Reviewer #2 (Remarks to the Author):

Dear Authors

I am satisfied with the thorough revision that has been made to address my points, as well as with the requested clarifications.

I am glad to support the publication of this study.

Kind regards

Response: We are grateful for all the very important comments the referee has raised and his/her time in improving our manuscript.

Reviewer #3 (Remarks to the Author):

The authors have done an excellent job of addressing all of the reviewer concerns. I believe this work is now suitable for publication in Nature Communications.

Response: Thanks a lot for the referee's positive comment. We also appreciate the referee for his/her time in reviewing our work.